# MULTI-SCALE ATTENTION FOR DIABETIC RETINOPATHY DETECTION IN RETINAL FUNDUS PHOTOGRAPHS

## ABSTRACT

The diagnosis and/or grading of diabetic retinopathy (DR) in the retina fundus has traditionally been done by physicians using manual procedures. However, there has been a significant demand for automated eye diagnostic and grading systems due to the constant rise in the number of persons with diabetes over the past few decades. An excellent diagnostic and predictive value for treatment planning exists with automatic DR grading based on retinal fundus pictures. With the majority of the current automated DR grading systems, it is exceedingly challenging to capture significant features because of the minor changes between severity levels. This paper presents a deep learning-based method for automatically assessing diabetic retinopathy in retina fundus pictures. This paper presents a deep learning-based method for automatically assessing diabetic retinopathy in retina fundus pictures. In order to increase the discriminative ability of the retrieved features, we implement a multi-scale attention mechanism within a deep convolutional neural network architecture in this research. Additionally, we provide a brand-new loss function termed modified grading loss that enhances the training convergence of the suggested strategy by taking into account the distance between various grades of distinct DR categories. The suggested technique is trained, validated, and tested using a dataset about diabetic retinopathy that is openly available. The experimental findings are presented to illustrate how well the suggested strategy competes.

## 1 INTRODUCTION

Diabetic Retinopathy(DR) is a disorder caused by excessive blood sugar levels that damages the rear of the eye (retina). It is a long-term microvascular issue brought on by uncontrolled diabetes mellitus (DM) and is one of the most significant consequences of type 2 diabetes (T2DM) Ali et al. (2016); Naserrudin et al. (2022). Diabetic retinopathy is classified into four types: no visible diabetic retinopathy (No DR), non-proliferative diabetic retinopathy (NPDR), proliferative diabetic retinopathy (PDR), and advanced diabetic eye disease (ADED). The multiple DR severity levels are broken down in Figure 1.

Physicians frequently use the international clinical DR severity scale developed by the American Academy of Ophthalmology (AAO) to classify patients as having non-proliferative diabetic retinopathy (NPDR), proliferative diabetic retinopathy (PDR), or maculopathy Ngah et al. (2020). It defined NPDR as the presence of any of the following disorders with no signs of proliferative retinopathy: micro-aneurysms, intra retinal hemorrhage, venous beading, or intra retinal microvascular abnormalities (IRMAs). Neovascularization, vitreous or pre-retinal hemorrhage, or both, were used to describe PDR. Following that, fundus images were labeled as having no DR, NPDR, PDR, advanced diabetic retinopathy (ADED), cataract, maculopathy, or glaucoma suspicious Ngah et al. (2020); Mallika et al. (2011). This classification assists in deciding when a referral is necessary, how frequently to monitor/screen patients, how to treat them, and other factors Saxena et al. (2020).

Manual eye screening for DR entails identifying this eye disorder through visual examination of the fundus, either through direct inspection (in-person dilated eye examinations) or through analysis of digital color fundus photographs of the retina. According to a number of studies (see e.g., Liesenfeld et al. (2000); Olson et al. (2003); Gangaputra et al. (2013)), fundus photography telemedicine has equivalent sensitivity and specificity as in-person screening for DR. Additionally, patients enjoy

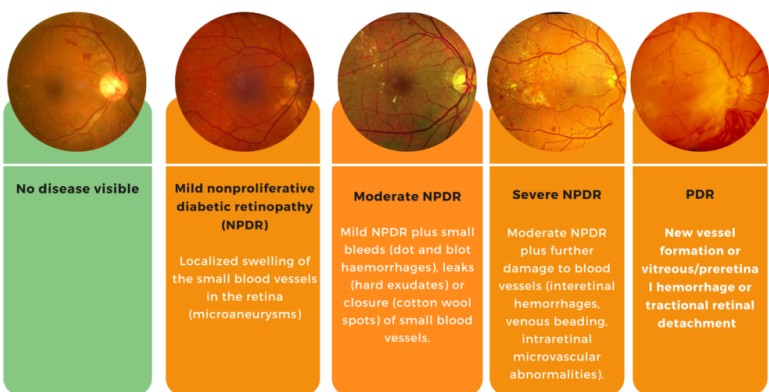

Figure 1: This is a figure. Schemes follow the same formatting. If there are multiple panels, they should be listed as: (**a**) Description of what is contained in the first panel. (**b**) Description of what is contained in the second panel. Figures should be placed in the main text near to the first time they are cited. A caption on a single line should be centered.Ophthalytics (2022)

using it and find it to be less expensive. Manually analyzing this image is hard and time-consuming, and the problem gets worse in rural areas where access to skilled medical professionals is limited. Therefore, there is a great demand for screening programs that are more effective, repeatable, and comprehensive. This will eliminate access barriers, enable early diagnosis and treatment, and improve patient outcomes.

Numerous research studies have been conducted in over the years to detect DR automatically, with a focus on feature extraction and binary-class prediction Baudoin et al. (1984); Frame et al. (1998); Niemeijer et al. (2009); Sinthanayothin et al. (2002); Niemeijer et al. (2005); Quellec et al. (2008); Abràmoff et al. (2009; 2013). These works have a few shortcomings but also some effectiveness. First of all, since they employed hand-crafted features, the characteristics that are extracted from images are prone to noise, exposure, and artifacts, to name a few. Second, feature placement and segmentation cannot be successfully included into the complete DR detection system. Additionally, just diagnosing to establish the presence or absence of DR rather than diagnosing the severity of DR could not adequately address real-world difficulties or assist physicians with their practice. Recently, deep learning techniques have found use in the processing of medical images, using Alyoubi et al. (2020) as an example.

In recent years, many studies on automatic DR grading/detection, including Gulshan et al. (2016); Abràmoff et al. (2016); Li et al. (2019); Zhao et al. (2019); Anoop et al. (2022); Bilal et al. (2021); Al-Antary & Arafa (2021); Zhao et al. (2020), utilized deep learning-based approaches, notably convolutional neural networks. These works make the best use of the autonomous feature extraction and excellent discriminability of deep learning (CNNs). In general, most of these studies utilized CNN for either binary classification Gulshan et al. (2016); Anoop et al. (2022) or multi-class prediction Abràmoff et al. (2016); Li et al. (2019); Zhao et al. (2019); Bilal et al. (2021); Al-Antary & Arafa (2021); Zhao et al. (2020). However, the approach ignores global information and loses crucial aspects Bello et al. (2019) due to the down-sampling operators in CNN (i.e., convolution and pooling processes). Existing DR detection systems based on pure convolutional neural networks suffer from this loss of semantic information.

This paper suggests to incorporate attention mechanisms within a deep convolutional neural architecture (Res-Net) for assessing diabetic retinopathy in retina fundus images. Motivated by the success of attention networks in machine translation Vaswani et al. (2017) and, more recently, computer vision tasks Li et al. (2018), this work embeds multi-scale attention mechanisms within the layers of a Res-Net56 architecture to improve classification accuracy for DR grading in fundus images. In contrast to earlier studies, the attention mechanisms used in this study are combined with a robust CNN architecture for automated DR grading, with a particular emphasis on the capacity of attention networks to automatically learn to focus on salient features at various phases of the feature extraction. Furthermore, we offer a novel loss function, dubbed modified grading loss, that improves

the proposed method's training convergence by accounting for the distance between distinct grades of various DR categories. The following are the distinctive contributions of this work:

1. We propose a novel attention-based automatic DR detection framework, thereby removing the worry of laborious and time-consuming manual analysis.

2. To facilitate early DR detection, we approach the problem of DR classification as a multi-class classification task.

3. To improves the proposed method's training convergence, we also propose a novel loss function, called modified grading loss.

## 2 BACKGROUND

### 2.1 DEEP RESIDUAL LEARNING

Naturally, deep networks incorporate many levels of features (low, mid, and high) and classifiers in an end-to-end multi-layer manner. By adding more stacked layers (i.e., increasing the network depth), these levels of features can be improved. However, as networks become deeper, accuracy becomes saturated and quickly deteriorates. According to Simonyan & Zisserman (2014); He et al. (2016), increasing the number of layers in a suitably deep model increases training error, proving that the problem of training accuracy degradation is not the result of "overfitting". To solve this issue of accuracy deterioration in "very deep" networks, He et al. presented the deep residual learning framework in He et al. (2016). In this method, the layers are allowed to fit a residual mapping rather than expecting that every few stacked layers directly fit a desired underlying mapping.

Technically, let consider $G(x)$ as an underlying mapping to be fitted by a few stacked layers (rather than the entire layers in the network), with x being the inputs to the first of these layers. Instead of expecting stacked layers to approach $G(x)$, the approach expressly allow them to approximate a residual function, $F := G(x) - x$. Thus, the initial function, $G(x)$ becomes $F(x) + x$. Despite the fact that both forms should be able to approximate the goal functions asymptotically (as intended), the ease of learning may differ. As demonstrated in Figure 2, feedforward neural networks with "shortcut connections" can be used to implement the formulation of $F(x) + x$.

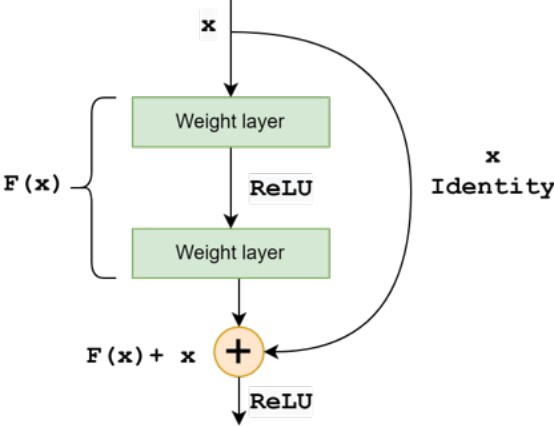

Figure 2: This is a figure. Schemes follow the same formatting.

Theoretically, a building block for deep residual learning is defined in He et al. (2016) as in Equation 1. In this case, the input and output vectors for the layers under consideration are **x** and y respectively. The residual mapping to be learned is represented by the function $F(\mathbf{x}, \{W_i\})$. For the two-layer example in Fig. 5, $F = W_2\psi(W_1\mathbf{x})$, where $\psi$ stands for ReLU [29] and the biases are included for notations simplification. By using a shortcut connection and element-wise addition, the operation $F + x$ is carried out.

$$\mathbf{y} = F(\mathbf{x}, \{W_i\}) + \mathbf{x} \tag{1}$$

In Equation 1, the dimensions of $x$ and $F$ must be the same (1). In the event that this is not the case (for instance, while switching the input/output channels), we can carry out a linear projection (refer to He et al. (2016) for details). It is important to state that, there are no additional parameters or computational complexity introduced by the shortcut connections in Eqn. (1). In addition to being desirable in practice, this is significant in our comparisons of plain and residual networks. To further enhance the discriminative capacity of deep convolutional neural networks, several methods have been suggested, while its still remain an open research problem.

## 2.2 ATTENTION MODELS FOR DEEP NETWORKS

One effective method that has been extensively adopted in earlier work is to use representations (such as those from the very top layer of a CNN) that compress visual information in an image down to the most prominent features. However, one possible disadvantage is that important semantic information are lost due to series of convolution and down-sampling operations. Lower-level representation can aid in the preservation of this information. Working with these features, however, demands a powerful method to direct the model to information relevant to the task at hand.

The presence of attention is one of the most perplexing aspects of the human visual system Xu et al. (2015). Bahdanau et al. first suggest the artificial attention mechanism for neural machine translation in Bahdanau et al. (2014). In recent years, attention models have excelled at a number of tasks involving computer vision and natural language processing Bahdanau et al. (2014); Xu et al. (2015); Chen et al. (2016); Vaswani et al. (2017); Dosovitskiy et al. (2020). Attention enables the model to concentrate on the most salient features as needed rather than compressing an entire image or sequence into a static representation Xu et al. (2015). This is especially helpful when an image has a lot of distraction. Stochastic attention and deterministic attention are two possible processes underlying the attention model. deterministic attention are often applied because they are differentiable, and are often referred to as "soft" attention.

Additive attention and dot-product (multiplicative) attention are the two most often employed attention functions. Formally, A query, a set of key-value pairs, and an output, all of which are vectors, can be expressed as an attention function by mapping them to the corresponding output. The output is generated as a weighted sum of the values, with the weight allocated to each value determined by the query's compatibility function with the relevant key Vaswani et al. (2017).

$$F_{att}(\mathbf{Q}, \mathbf{K}, \mathbf{V}) = softmax\left(\frac{\mathbf{Q}\mathbf{K}^T}{\sqrt{d_k}}\right)\mathbf{V} \tag{2}$$

In practice, the attention function, $F_{att}$ is compute same time on a collection of queries pack into a matrix $\mathbf{Q}$. Also, the matrices $\mathbf{K}$ and $\mathbf{V}$ include both the keys and the values. In Eqn 2, we provide the mathematical expression for a dot-product attention as given in Vaswani et al. (2017).

## 3 METHOD

### 3.1 PROPOSED MULTI-SCALE ATTENTION NETWORK FOR DR GRADING (MAN-DR)

In this paper, we proposed the Multi-scale Attention Network for DR grading (MAN-DR) in fundus pictures. The "extremely deep" convolutional neural structure of the RAN-DR convolutional network employs a mixed attention approach. As a baseline for this work, we adopt Chen et al. (2016) , and the internal structure of the suggested RAN-DR is shown in Fig. 3. The development of the soft attention structure in our MAN-DR was motivated by the fact that there are only marginal differences across the different severity levels of DR and the need to merge the multi-scale features from the CNNs.

A type of deep network layout referred to as "share-net" Farabet et al. (2012); Papandreou et al. (2015) that resize the input image to several scales and run each scale through a separate deep network is employed. The final prediction is then computed based on the fusion of the obtained

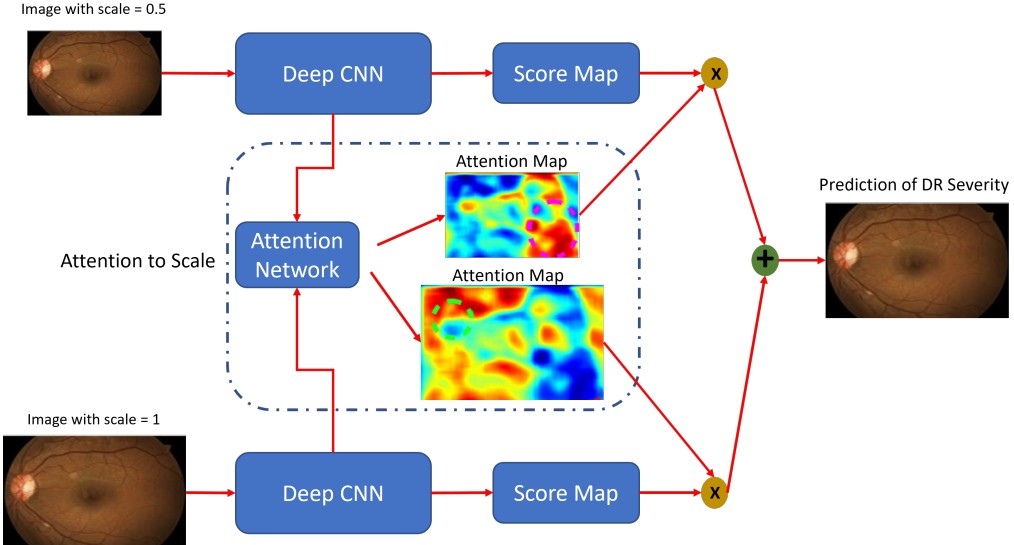

Figure 3: An illustration of the internal structure of the proposed network architecture Wang et al. (2017).

multi-scale features. To merge the multi-scale features for our proposed model, We suggest an attention model that learns to weight the multi-scale features.

Assume an input image is resized to many scales $s \in 1, ..., S$ using share-net. Each scale is run through a modified version of DeepLab Papandreou et al. (2015) (Replacing the CNNs with Res-Net56, while the FCN weights are shared across all scales) to provide a score map for scale $s$, indicated as $f_{k,c}^s$ where k spans all spatial coordinates (because it is completely convolutional) and $c \in 1, ..., C$ where C is the number of classes of interest. Bilinear interpolation is used to enlarge the score maps $f_{k,c}^s$ to the same resolution (with regard to the finest scale).

For all scales, we denote $\widetilde{J}_{k,c} J_{k,c}$ as the weighted sum of score maps at $(i, c)$:

$$\widetilde{J}_{k,c} = \sum_{s=1}^{S} w_k^s \cdot f_{k,c}^s \tag{3}$$

$w_k^s$ can be expressed as given in Chen et al. (2016) as:

$$w_k^s = \sum_{i=1}^{s} \frac{\exp(h_s^k)}{\sum_{i=1}^{s} exp(h_s^k)} \tag{4}$$

where $h_s^k$ is the attention model's score map (i.e., the last layer output before SoftMax) at location $k$ for scale $s$. It's worth noting that wsi is shared across all channels.

### 3.1.1 PROPOSED LOSS FUNCTION

As part of the optimization algorithm, the error of the current state of the model must be estimated repeatedly. However, traditional loss functions are only capable of breaking down a multi-class classification into a number of binary classifications. In these typical loss functions, the distance between various classes is not taken into account. To address this problem "grading loss" function was proposed by Zhao et al. (2019) which adds weights to the softmax function in order to decrease the loss-accuracy discrepancy and achieve improved convergence. This approach however adopt a distance-based weight function, which is a weighted softmax. In this work we intoduced a normalization factor into the error function.

## 3.2 DATASET

We utilized publicly available datasets for Kaggle competition (Kaggle EyePACS) Kaggle (2015) for training, validation, and evaluation of the proposed algorithm. The Kaggle EyePACS datasets include 88,702 anonymized images of the retinal fundus obtained for diabetes screening. The dataset includes high-resolution retina images captured under various imaging conditions. For each subject, a left and right field is provided. In the dataset folder, containing train and test folders, images are identified by a subject id and are either labeled as left or right (for example, '15_left.jpeg' is the left eye of patient id 1). On a scale of 0 to 4, a clinician rated the presence of diabetic retinopathy in each image. A .csv is provided with the dataset, with two columns (image and level) for and image path location (image) and severity (level). For the severity level, the number simply implies: 0 – No DR, 1 – Mild DR, 2 – Moderate DR, 3 – Severe DR, and 4 – Proliferative DR. Figure 2 (a) shows some samples of the training set with data augmentation, while Figure 1 (b) shows the validation set. The training set contains 25810 level 0 images, 2443 level 1 images, 5292 level 2 images, 873 level 3 images, and 708 level 4 images. In Figure 3, we provide the plot of the dataset distribution.

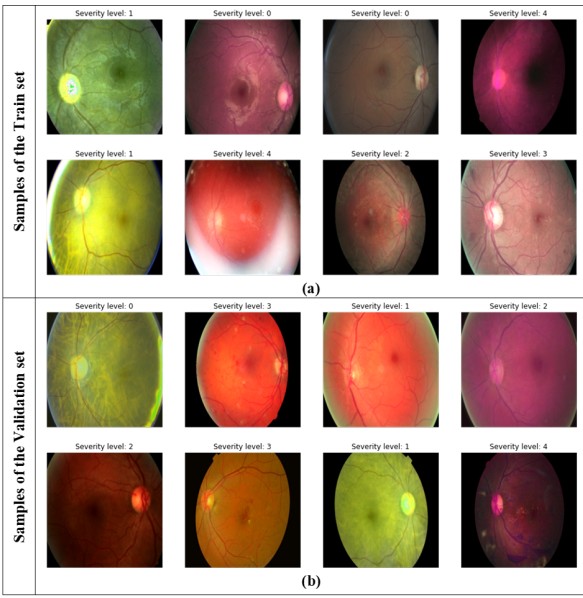

Figure 4: Samples from the employed dataset: (**a**) Some data instances from the training set (**b**) Some data instances from the validation set

### 3.2.1 DATA PREPROCESSING

The dataset available for this study was unbalanced, we however, chose to divide it into the following parts: 312 images per class make up the 1560 images used for validation, while the 33566 images used for training (with 25496, 2132, 4980, 561, 397 images per class, corresponding to classes 0 to 4). This split was made to ensure that both eyes of each subject in the database were in the same set. We performed data augmentation using rotations from 0 to 360 degrees, zooming in and out from 0 to 20 pixels, and performing horizontal and vertical flips in order to balance the training set and increase the number of images for training. All of the training set's images underwent random augmentation, and for classes 1, 2, 3, and 4, identical images underwent repeated augmentation with different augmentation parameters. 127480 images made up the resulting balanced training data set. In Figure 3, we provide some samples of the training and validation set for this study.

## 4 EXPERIMENT

In this section, we conduct experiments to evaluate the accuracy and robustness of our MAN-DR. We compare our model to the work of BIRA-NET [34] and SEA-NET [23], where attention mech-

anisms are also employed. Ablation experiments are used to analyze the performance of different combinations among distinct sections in order to investigate the usefulness of various modules in the proposed MAN-DR.

**Implementation detail:** Tensorflow and the Keras API are used to implement the proposed model. The training was accelerated by a single NVIDIA Tesla P100 GPU with 16GB VRAM, and the model was trained using a stochastic gradient descent (SGD) optimizer with a momentum of 0.9 and batch size of 32. The $l2$ regularization is applied to weights with a weight decay factor of 5e-7 and an initial learning rate of 0.05. In Fig. 5, we provide the plot of epoch against accurracy for both training and validation.

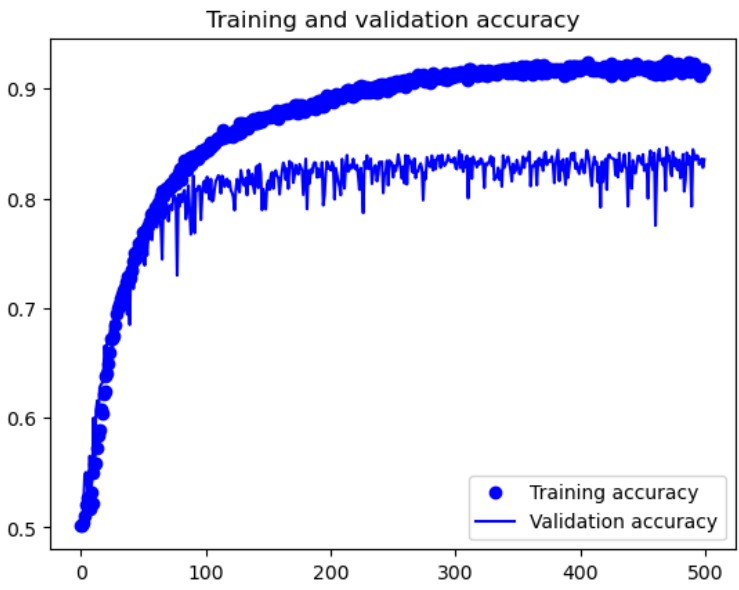

Figure 5: Plot of training and validation accuracy

**Performance measure:** We construct the confusion matrix, in which the number of predictions in each class are shown, to measure performance on the multi-class classification problem. The proportion of the diagonal elements, which reflect the number of points for which the predicted label is equal to the true label, can be used to determine the average of classification accuracy (ACA) based on the confusion matrix. Receiver operating characteristics (ROC) Hajian-Tilaki (2013), which reflect the true positive rate (TPR) and the false positive rate (FPR) at different threshold settings, are calculated to assess the diagnostic performance of the suggested approach. AUC (Area Under Curve), which is based on ROC, is calculated for comparison.

Figure 6 depicts the confusion matrix for the proposed MAN-DR, with the horizontal axis representing the anticipated classes and the vertical axis representing the genuine classes. This confusion matrix is not normalize in other to reveal the true values of correctly classify and wrongly classify instances.

## 4.1 Discussion

Table 1 displays the experimental outcomes of all methods on the testing data. Even without the proposed loss function, the suggested framework outperforms competing techniques in AUC. The suggest approach shown great deal of improvements as to when the share-net architecture is employed with attention. Using the proposed error function, model convergence rate increases. Except for class 3 qnd 4, which is generally classified into class 0,1, or 2, each class in the confusion matrix is most likely to be forecasted into the correct class. It is obvious that class 3, 4 is the most difficult to distinguish, whereas normal (class 0) is the easiest.

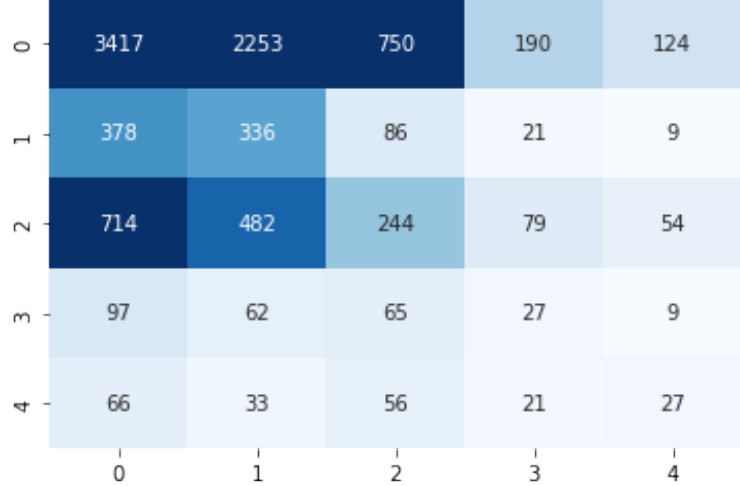

Figure 6: Classification Confusion matrix for the proposed MAN_DR

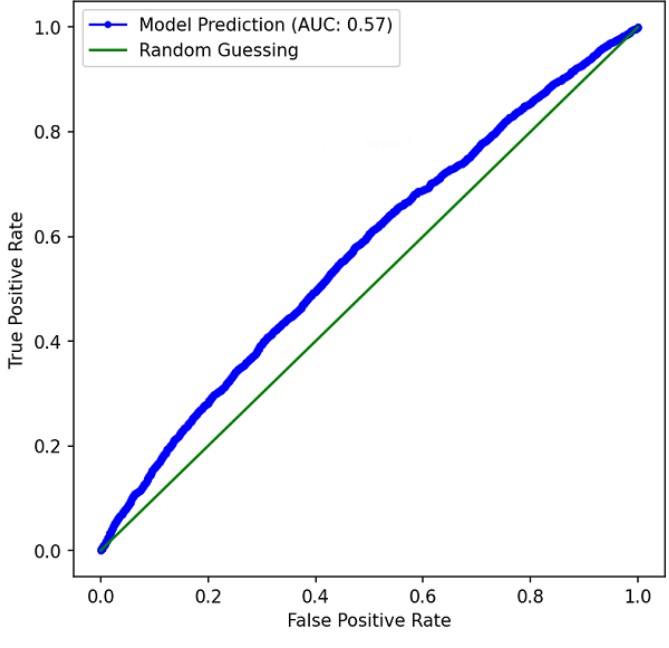

Figure 7: Receiver operating characteristics computed for the proposed model

## 5 CONCLUSION

The need for regular eye screening for diabetes patients has proven to be an effective measure to prevent DR-related vision impairments. Owing to sustained increment in number of individuals living with diabetes globally, there have been high demanded for automated DR grading. This research presents a multi-scale attention-based deep learning architecture for diabetic retinopathy grading, with a share-net structure implemented for learning multi-scale high discrimantive feature is applied. Furthermore, the proposed grading loss function contributes to the proposed approach's substantially improved convergence. The ablation analyses reveal that the proposed components significantly increase classification performance. The suggested MAN-DR is competitive with state-of-the-art approaches, as demonstrated by the experimental findings.

ACKNOWLEDGMENTS

This work is supported by the YUTP Fundamental Research Grant.

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

## A  APPENDIX

You may include other additional sections here.

