# OpenReview forum: "Multi-scale Attention for Diabetic Retinopathy Detection in Retinal Fundus Images"
_ICLR.cc/2023/Conference — Submitted to ICLR 2023_

### Official Review · Reviewer_mS2T · 2022-10-24

**Confidence:** 5
**Correctness:** 3
**Technical Novelty And Significance:** 2
**Empirical Novelty And Significance:** 2
**Recommendation:** 3

**Clarity, Quality, Novelty And Reproducibility:**

The clarify of this work is not bad and the writing of this work is easy to follow.
However, the quality and the originality of this work is not acceptable. This work has a limited technical novelties. Moreover, the experiments are not convinced, and the authors mainly focus on the results of the developed method.


**Strength And Weaknesses:**

Strengths:
1.	This work develops a CNN based on a multi-scale attention for DR grading.
2.	The experimental results verify the effectiveness of the developed method.
Weaknesses:
1.	This work has limited technical novelties. The multiscale attention strategy is to simply learn attention maps at different scales.  Such technical novelties are not enough for publication in ICLR
2.	The experimental results are not convinced. It seems that the authors only report the results of the developed method. It neglects the comparisons with state-of-the-art methods.
3.	In the experiments, the authors do not provide any ablation study experiments.


**Summary Of The Paper:**

This work develops a deep network for assessing diabetic retinopathy (DR) in retina fundus images by implementing a multi-scale attention mechanism, and a brand-new loss function.
Experimental results show that the effectiveness of the developed method.


**Summary Of The Review:**

1.	The technical novelties of this work are not enough for publication in ICLR.
2.	The experiments are not convinced. The authors do not provide comparisons against state-of-the-art methods and any ablation study experiments.
3. 	Writing issues: The sentence “This paper presents a deep learning-based method for automatically assessing diabetic retinopathy in retina fundus pictures.” appears two times in the Abstract.

---

> ### Author Response · Authors · 2022-11-26
> **Response to Reviewer mS2T**
>
> Thank you so much for your effort towards making our work better.  And I apologise for responding late. We provide comparison with SOTA approaches and the details of the comparison were contained in the table 1 cited but was missing do to  some mistakes during the latex compilation and hurry to meet the submission deadline.
> About novelty, we think our unique contribution was stated in the abstract. The aim was to utilised an attention based feature fusion mechanism within a  CNN architecture in other to improve the discriminative capability. Unlike other works, we utilised multi-stage attention as opposed single-stage attention which were common in other works in this area.

---

### Official Review · Reviewer_5o7H · 2022-10-24

**Confidence:** 5
**Correctness:** 2
**Technical Novelty And Significance:** 1
**Empirical Novelty And Significance:** 1
**Recommendation:** 3

**Clarity, Quality, Novelty And Reproducibility:**

The paper is almost clear and reproducible. However, the novelty and the scientific quality in terms of contributions are notably limited.


**Strength And Weaknesses:**

Strength:
-The paper is well-written and easy to understand.
-Developing applications of deep learning in other domains especially healthcare is highly important and requires major attention from the community.




Weaknesses:
MAJOR:
-A major shortcoming of the paper is the lack of novelty. The attention mechanism, network architectures, the idea of passing different scales of the same input to different networks, and the grading loss function are cited from previous works and the current work seems as an adaptation of a previously developed model for the special tasks of DR classification. Further, the task itself is also a simple classification problem that does not incur novel challenges. A very similar paper to the current work is also published: “Al-Antary, Mohammad T., and Yasmine Arafa. "Multi-scale attention network for diabetic retinopathy classification." IEEE Access 9 (2021): 54190-54200.”.
-The reasoning for developing soft attention is vague and requires further clarification. Perhaps, conducting a relevant experiment can be beneficial to clarify its usefulness.
-The description of the grading loss function is vague and requires a formal formulation to clarify the concept.
-Only a single dataset is used for the evaluation. This limits the comprehensiveness of the validations.
-How is the ROC curve for the 4-class classification problem computed?


MINOR:

-“(DR)” in the first line of the introduction requires a space.
-“This is a figure” in Figure 2 is redundant.
-Section 3.1.1: “this approach adopt” -> “this approach adopts”
-Section 3.1.1: “In this work we intoduced” requires a comma after work and also there is a typo in introduced.
-Section 4.1: “the suggest approach” -> “the suggested approach”
-Section 4.1: “shown” -> “shows”
-Section 4.1: “class 3 qnd 4” -> “class 3 and 4”
-Section 4.1: “class 3, 4” -> “classes 3 and 4”


**Summary Of The Paper:**

The paper presents a deep-learning method for the automatic assessment of diabetic retinopathy in retina fundus pictures. A multi-scale attention mechanism is developed to enhance the discriminative power of the model. In contrast to the previous methods, the attention mechanisms used in this work are combined with the network architecture to automatically learn to focus on salient features at various phases of the feature extraction. Furthermore, a novel loss function, named modified grading loss, is proposed to improve the training convergence by accounting for the distance between distinct grades
of various DR categories.


**Summary Of The Review:**

The paper combines several ideas from previous works including a multi-scale attention mechanism and grading loss function to improve the classification of DR in retina images. The methodology lacks novelty and the problem itself is not challenging enough. Furthermore, the experimental setup is simple and does not present enough evaluations and ablation studies to validate the claims in the paper including the necessity of soft or multi-scale attention and the grading loss.

---

> ### Comment · Reviewer_5o7H · 2022-11-24
> **Updated comments**
>
> Since the authors have not provided any response to the comments, I keep the previous rating based on the improper problem formulation, lack of novelty, and ambiguities in the methodology discussed in Section 2.  I encourage the authors to consider the comments and perform a major revision.

---

### Official Review · Reviewer_zeSz · 2022-10-25

**Confidence:** 5
**Correctness:** 1
**Technical Novelty And Significance:** 1
**Empirical Novelty And Significance:** Not applicable
**Recommendation:** 1

**Clarity, Quality, Novelty And Reproducibility:**

The novelty, quality and reproducibility of the work is limited due to some missing results and other omissions specified in the previous comment.

**Strength And Weaknesses:**

The paper addresses an important problem of diabetic retinopathy (DR) grading, what's important in encouraging further improvements and automation of medical procedures. The background of the problem is clearly explained along with existing methods used for the prediction of the DR disorder.

However, the work suffers from some significant omissions that affect its quality:
1) As specified in the experiments section, other techniques for DR grading using attention mechanisms have been proposed. It's not clear how the method differs from them.
2) Comparison of results is missing - there is no Table 1 that is referred to in the text.
3) Results are not satisfactory, the performance of the model is quite poor, what hasn't been discussed. A details explanation of results, ideas for improvements and future work would be beneficial.
4) It's mentioned that the used dataset was imbalanced, what can be also seen in the presented confusion matrix. Majority of samples were classified as category 0. Have you tried any techniques to address this issue, such as reweighing/over/undersampling, etc.?
5) Weighted softmax has been used in other studies before. How exactly does the new loss function differ from that? It's not clear.
6) Selection of specific neural networks hasn't been explained. It would be interesting to compare different topologies.
7) Background section contains explanation of well known techniques. It's sufficient to refer to them instead of introducing them in details. Similarly, attention mechanism is known, so there is no need for such detailed introduction of this technique.

**Summary Of The Paper:**

The work proposes to combine a multi-scale attention mechanism with a deep convolutional neural network to improve discriminative abilities of the features used for diabetic retinopathy grading. The modified training loss is used for improving model convergence by comparing various grades of the disorder. The experiments are performed on the publicly available dataset to show advantages of the method over other attention-based techniques.

**Summary Of The Review:**

The presented paper looks incomplete. Some additional work should be done to improve its quality, provide more detailed analysis of results and justify the novelty of the method.

---

> ### Comment · Reviewer_zeSz · 2022-11-26
> **Update**
>
> There was no update to the work, so my rating remains unchanged.

---

### Decision · Program_Chairs · 2023-01-20

**Decision:**

Reject

**Justification For Why Not Higher Score:**

This is submission with some important results missing.

**Justification For Why Not Lower Score:**

N/A

**Metareview: Summary, Strengths And Weaknesses:**

This paper presents an approach for DR grading. But the authors failed to include some important results during the paper submission.  The reviewers also raise many concerns including the technical novelty etc. The authors did not respond to many of reviewers' comments and therefore the reviewers remain their opinions to reject the paper.

**Summary Of Ac-Reviewer Meeting:**

NA.